# Molecular Alterations in Dog Pheochromocytomas and Paragangliomas

**DOI:** 10.3390/cancers11050607

**Published:** 2019-04-30

**Authors:** Esther Korpershoek, Daphne A. E. R. Dieduksman, Guy C. M. Grinwis, Michael J. Day, Claudia E. Reusch, Monika Hilbe, Federico Fracassi, Niels M. G. Krol, André G. Uitterlinden, Annelies de Klein, Bert Eussen, Hans Stoop, Ronald R. de Krijger, Sara Galac, Winand N. M. Dinjens

**Affiliations:** 1Department of Pathology, Erasmus MC Cancer Institute, University Medical Center, 3015 GD Rotterdam, The Netherlands; daphne93@live.nl (D.A.E.R.D.); a.krol@erasmusmc.nl (N.M.G.K.); j.stoop@erasmusmc.nl (H.S.); w.dinjens@erasmusmc.nl (W.N.M.D.); 2Department of Pathobiology, Faculty of Veterinary Medicine, Utrecht University, 3584 CL Utrecht, The Netherlands; G.C.M.Grinwis@uu.nl; 3School of Veterinary and Life Sciences, Murdoch University, Murdoch 6150, WA, Australia; profmjday@gmail.com; 4Clinic for Small Animal Internal Medicine, University of Zurich, 8057 Zurich, Switzerland; creusch@vetclinics.uzh.ch; 5Institute of Veterinary Pathology, University of Zurich, 8057 Zurich, Switzerland; hilbe@vetpath.uzh.ch; 6Department of Veterinary Medical Science, University of Bologna, 40064 Ozzano dell’Emilia, Italy; federico.fracassi@unibo.it; 7Genetic Laboratory, Erasmus MC University Medical Center, 3015 GD Rotterdam, The Netherlands; a.g.uitterlinden@erasmusmc.nl; 8Department of Clinical Genetics, Erasmus MC University Medical Center, 3015 GD Rotterdam, The Netherlands; a.deklein@erasmusmc.nl (A.d.K.); h.eussen@erasmusmc.nl (B.E.); 9Department of Pathology University Medical Center/Princess Maxima Center for Pediatric Oncology, 3584 CS Utrecht, The Netherlands; R.R.deKrijger@umcutrecht.nl; 10Department of Clinical Sciences of Companion Animals, Faculty of Veterinary Medicine, Utrecht University, 3584 CL Utrecht, The Netherlands; S.Galac@uu.nl

**Keywords:** dog, pheochromocytoma, paraganglioma, *SDHB*, *SDHD*, mutation, chromosomal alteration, comparative genomics

## Abstract

Recently, genetic alterations in the genes encoding succinate dehydrogenase subunit B and D (*SDHB* and *SDHD*) were identified in pet dogs that presented with spontaneously arising pheochromocytomas (PCC) and paragangliomas (PGL; together PPGL), suggesting dogs might be an interesting comparative model for the study of human PPGL. To study whether canine PPGL resembled human PPGL, we investigated a series of 50 canine PPGLs by immunohistochemistry to determine the expression of synaptophysin (SYP), tyrosine hydroxylase (TH) and succinate dehydrogenase subunit A (SDHA) and B (SDHB). In parallel, 25 canine PPGLs were screened for mutations in *SDHB* and *SDHD* by Sanger sequencing. To detect large chromosomal alterations, single nucleotide polymorphism (SNP) arrays were performed for 11 PPGLs, including cases for which fresh frozen tissue was available. The immunohistochemical markers stained positive in the majority of canine PPGLs. Genetic screening of the canine tumors revealed the previously described variants in four cases; *SDHB* p.Arg38Gln (*n* = 1) and *SDHD* p.Lys122Arg (*n* = 3). Furthermore, the SNP arrays revealed large chromosomal alterations of which the loss of chromosome 5, partly homologous to human chromosome 1p and chromosome 11, was the most frequent finding (100% of the six cases with chromosomal alterations). In conclusion, canine and human PPGLs show similar genomic alterations, suggestive of common interspecies PPGL-related pathways.

## 1. Introduction

Pheochromocytomas (PCCs) and paragangliomas (PGLs; together PPGLs) are tumors arising from chromaffin tissue inside (PCC) and outside the adrenal glands (PGL). These tumors occur in the context of several hereditary syndromes, such as Von Hippel Lindau (VHL), Multiple Endocrine Neoplasia type 2, Neurofibromatosis type 1, and the PCC-PGL syndrome, with underlying germline mutations in the *VHL*, rearranged during transfection (*RET*), neurofibromin 1 (*NF1*), and the *SDH*-genes, respectively [1]. Both germline and somatic mutations can be found in more than 20 genes [2,3]. Although approximately 10% of PPGL patients, in general, will present with (distant) metastases, this frequency is much higher in patients with succinate dehydrogenase subunit B (*SDHB*) germline mutations. During follow-up, more than 35% of *SDHB* patients will present with PPGL metastases [4,5]. 

In an effort to unravel the mechanisms behind malignant behavior of PPGLs, and more specifically the metastatic behavior of *SDHB*-related tumors, several attempts have been made to generate knock-out mouse models. These models either proved lethal during embryogenesis or the mice did not develop PPGL or other *SDH*-related tumors [6]. The only mouse models that presented with high frequencies of PCCs were based on conditional homozygous inactivation of *Pten* and heterozygous conventional inactivation of *Nf1* [7]. However, human PPGLs have never been associated with phosphatase and tensin homolog (*PTEN*) mutations [8]. In addition, *NF1*-related PPGLs are relatively benign with metastatic behavior seen in fewer than 10% of cases [9]. So, although these PPGL mouse models are interesting, there remains a need for an appropriate animal model to study *SDH*-related PPGL. 

Recently, Holt et al. reported genetic screening of eight canine PPGLs and identified four genetic variants that might be potentially pathogenic; one in *SDHB* (p.Arg38Gln) and three in succinate dehydrogenase subunit D (*SDHD*; p.Lys122Arg), of which one was somatic [10]. Because these alterations occurred in highly conserved amino acids, the authors assumed that the alterations were likely pathogenic. In fact, the somatic event suggests that at least the SDHD p.Lys122Arg amino acid change is likely pathogenic, while the *SDHB* alteration remains of unknown significance. Since this is the first animal model that presents spontaneously with PPGL that might be related to *SDHB* and *SDHD* mutations, we have investigated a relatively large series of canine PPGLs by Sanger sequencing for mutations in these genes. In addition, immunohistochemistry was performed for tyrosine hydroxylase, synaptophysin, SDHB, and succinate dehydrogenase subunit A (SDHA), and SNP arrays to identify chromosomal alterations. 

## 2. Results

### 2.1. Clinical Findings

The canine PPGL immunohistochemistry series included 32 PCCs and 18 PGLs. The average age of the dogs at diagnosis was 11 years for PCCs (ranging from 4 to 16 years) and 9 years for PGLs (ranging from 2 to 11 years). In the PCC group, the distribution of males and females was almost identical (53% male), while the PGL group included more tumors from males (72%). Metastatic behavior was reported in 3% of the dogs with PCCs (*n* = 1) and 11% of the dogs with PGL (*n* = 2). 

### 2.2. Genetic Analyses

Sanger sequencing results are shown in Table 1. Twenty-one PPGLs (20 PCCs and one PGL) with sufficient DNA quality and positive synaptophysin (SYP) or tyrosine hydroxylase (TH) immunohistochemical labeling, were screened for mutations in the *SDHB* and *SDHD* genes. If a tumor had a non-synonymous variant, the presence of this variant was also investigated in corresponding germline DNA. 

Three PCCs showed an SDHD (XM_536573) c.365A>G; p.Lys122Arg alteration. In PCC6 and PCC46 the variant was homozygous, while in PCC23 the alteration was heterozygous. Germline DNA was only available for PCC6 and showed the *SDHD* variant in a heterozygous fashion, indicating loss of the wild type (WT) *SDHD* allele in the tumor (Figure 2), which was also confirmed by the loss of heterozygosity analyses for microsatellite markers flanking the *SDHD* gene (Figure 1B). PCC19 showed the previously described *SDHB* (NM_001252217) c.113G>A; p.Arg38Gln alteration, which appeared homozygous in the tumor and corresponding germline DNA of this dog. (Figure 1A)

In addition, genetic screening for *SDHA* mutations was only performed in SDHA immunohistochemically-negative tumor PCC1 (see below Figure 2). Due to the relatively poor quality of the sample it was not possible to obtain *SDHA* DNA sequences of sufficient quality to analyze. 

### 2.3. Immunohistochemistry 

SYP was positive in 86% of PCCs and 71% of PGLs, while TH was positive in 74% of PCCs and 35% of PGLs. Results of the immunohistochemistry of all tumors are listed in Appendix A and illustrated in Figure 1. SDHB immunohistochemistry was performed on all 50 canine PPGLs. PCC1 and PCC6 showed heterogeneous labeling for SDHB, with foci of tumor cells that were immunohistochemically negative for SDHB and areas that were weakly positive. The SDHA labeling for PCC1 also appeared to be negative for the tumor cells in this PCC (Figure 2), while all other tumors were positive. All other PPGLs, as well as the positive control tissues (normal dog adrenals), were immunohistochemically positive, although they did not show the typical granular labeling pattern. 

### 2.4. SNP Arrays

Chromosomal alterations were investigated for 11 dog PPGLs by SNP arrays. From those, analysis was not possible for two samples due to high background noise. In addition, three tumors did not show any chromosomal alterations. The chromosomal alterations of the six remaining tumors are listed in Table 2. Furthermore, an illustrative single nucleotide polymorphism (SNP) array result of case PCC20 (logR ratios and b-allele frequencies (BAF)) is depicted in Figure 1C, showing loss of chromosomes 5, 17, 23, 26, 30, and 34. The most frequent genomic alteration was loss of chromosome 5, which occurred in all six dog PPGLs (100%), followed by loss of chromosome 26 in 5/6 dog PPGLs (83%). Chromosome 5 (CanFam3.1) is, for a large part, syntenic to two areas of human chromosome 1 (GRCh38.p3), and to a part of human chromosome 11, including the *SDHD* region, respectively (see Figure 3, Appendix A). Genomic locations of *SDHA*, *SDHB*, *SDHC,* and *SDHD* are shown in Appendix A. 

## 3. Discussion

Currently, there is still no curative therapy for patients with metastatic PPGL. In general, malignant behavior occurs in 10% of patients with PPGL. However, patients with *SDHB* germline mutations have a much higher chance of developing distant metastases [11]. Investigating animal models of PPGL could lead to the development and testing of therapies for humans with metastatic PPGL. Thus far, the only animal model that presents with metastatic PCC is a *Pten* KO mouse [12,13]. Since *PTEN* mutations do not play a role in the pathogenesis of human (malignant) PPGL, such models are not the most suitable for the study of human malignant PPGL [8]. A recent study reported that dogs presenting spontaneously with PPGL had potential pathogenic genetic alterations in *SDHB* and *SDHD* [10]. To investigate these results in an independent and larger series we have screened 25 canine PPGLs for mutations and identified one *SDHB* and three non-synonymous *SDHD* genetic alterations, both of which have been described previously [10]. 

The SDHD p.Lys122Arg variant was identified in three canine PPGLs; once in a heterozygous (PCC23) and twice in a homozygous fashion (PCC6 and PCC46). The corresponding germline DNA from PCC6 showed the variant to be present in a heterozygous fashion, indicating loss of the wild type *SDHD* allele in the tumor DNA. This finding was confirmed by the loss of heterozygosity (LOH) analysis, using microsatellites flanking the *SDHD* gene. Although there was no germline DNA available for PCC46, the homozygous expression of the SDHD p.Lys122Arg variant in the tumor could be explained by the potential loss of the *SDHD* wild type, since the SNP array results showed loss of chromosome 5, which includes the *SDHD* gene. However, this is only speculation, since we cannot confirm the possibility that the *SDHD* variant was heterozygous in the germline DNA, and could also be present as a homozygous SDHD p.Lys122Arg or homozygous wild type. 

In addition, another previously described non-synonymous *SDHB* variant [10] was detected in PCC19. This tumor showed the SDHB p.Arg38Gln variant in a homozygous fashion, which was also homozygous in DNA isolated from normal tissue from the same dog. Although the SDHB p.Arg38 is a highly conserved amino acid throughout many species, the fruit fly has a Gln at position 38. Since the variant was already present in a homozygous fashion in the germline, no loss of the *SDHB* locus was seen in the SNP array results. As the p.Arg38Gln amino acid change is probably not deleterious to the function of the protein, we regard this variant as likely benign. However, this should be further investigated functionally.

Sanger sequencing screening of this tumor did not reveal any novel mutations in *SDHB* and *SDHD*. However, mutations could have been missed due to the poor DNA quality, resulting in sequences that could not be analyzed, and due to other driving mechanisms, such as promoter-methylation or large (exon) deletions, which are not detectable by Sanger sequencing [14,15,16].

The SDHA and SDHB immunohistochemistry appeared to be positive in almost all investigated tumors, with the exception of PCC1 and PCC6. PCC1 also was immunohistochemically negative for SDHA. However, since the positive control tissues (normal dog adrenals) showed homogeneous and not granular cytoplasmic labeling, we suspect that the antibody is not suitable for screening for SDH mutations, as in human PPGL [17,18], and, therefore, no conclusions can be drawn from the SDHA and SDHB immunohistochemistry. In addition, the negative/weak positive SDHA and SDHB labeling could also be due to technical limitations, such as fixation artifacts.

The SNP array results showed large chromosomal alterations in six of the nine PPGLs with informative SNP arrays. The fact that three tumors did not show chromosomal aberrations was most likely due to the low neoplastic cell content of these frozen tissue samples, from which DNA was isolated. From the six tumors that showed chromosomal alterations, loss of chromosome 5 was the most frequent alteration (100%). Canine chromosome 5 shows homology with regions of human chromosome 1p and 11q. Many studies have shown that loss of chromosome 1p and chromosome 11 are frequent events in the pathogenesis of human PPGL [3,19,20]. Our results suggest that there are common genes, located in these homologous chromosomal areas, that might contribute to the pathogenesis of both human and canine PPGL. 

## 4. Materials and Methods 

### 4.1. Patients and Sample Selection 

In total, we collected 50 dog PPGLs from 45 dogs (including one case with bilateral tumors), which comprised of 44 formalin-fixed paraffin wax-embedded (FFPE) blocks and six fresh frozen samples (FF), from contributing Veterinary faculties from the University of Bristol, Bristol, United Kingdom (*n* = 25), Utrecht University, Utrecht, the Netherlands (*n* = 12), University of Zurich, Zurich, Switzerland (*n* = 8), and University of Bologna, Bologna, Italy (*n* = 5). In addition, we also collected three normal canine adrenal glands provided by the Veterinary Faculty of Utrecht University, to be used as positive controls for the immunohistochemistry. In addition, from two dogs, corresponding germline DNA was available (PCC6 and PCC19). All tissue samples were collected during surgery or necropsy examination from pet dogs suffering from PPGL. The owners of the dogs had given permission for the tissues to be used for research purposes. All clinical characteristics including age, breed, and gender are listed in Appendix A. PPGLs were only considered malignant if distant metastases were present, as for human PPGL. A summary of the current study is shown in Appendix A. 

### 4.2. Immunohistochemistry

Immunohistochemistry was performed for all canine samples to confirm the diagnosis of PPGL using markers for synaptophysin (SYP) and tyrosine hydroxylase (TH). SYP and TH were evaluated by Esther Korpershoek and Daphne Dieduksman and scored as positive if there was a weak to strong specific expression in the cytoplasm of all tumor cells. SYP was labeled using the Ventana Benchmark ULTRA automated immunohistochemistry stainer (details available on request), while TH immunohistochemistry was performed as previously described [13]. Since in human PPGL, negative SDHB immunohistochemistry reliably identifies tumors with mutations in *SDHA*, *SDHB*, *SDHC*, or *SDHD*, we also investigated the immunohistochemical expression of SDHB in the canine PPGLs [21]. The peptide to which this SDHB antibody was generated is 99% homologous to the canine peptide; only one of 108 amino acid residues is different. The series of canine PPGLs was also labeled for SDHA, as described previously [17]. Normal dog adrenal glands were used as positive controls for immunohistochemistry. The immunohistochemistry was performed on 5 μm sections using the Ventana automated stainer as described previously [18]. SDHB and SDHA expression was scored as positive if strong expression was observed in all cells, while a tumor was considered as negative if the labeling of the tumor cells was negative or weakly positive compared with the positive granular labeling present in the surrounding endothelial cells, which serve as internal positive controls [17,21].

### 4.3. Genetic Screening

In total, 24 PPGLs (Appendix A) that expressed SYP and/or TH immunohistochemically were selected for the genetic screening and one TH/SYP negative PCC was also studied. For the FFPE samples, DNA isolation was performed by manual microdissection, to ensure that the DNA was derived from a high percentage of neoplastic cells. DNA isolation was performed with the DNaesy kit (#69504, Qiagen, Venlo, the Netherlands) according to the manufacturer’s protocol. DNA concentrations were measured with the Qubit® dsDNA HS BR assay kit (#Q32850, Thermo Fisher, Waltham, MA, USA) according to the manufacturer’s instructions. For the generation of the primers, human *SDHA* (NM_004168), *SDHB* (NM_003000), and *SDHD* (MN_003002) mRNA sequences were aligned with the dog genome (CanFam3.1/canFam3) to identify the exact location and sequences of the genes and exons, enabling the generation of dog-specific primers covering all of the coding sequences. PCR was performed with KAPA2G Hotstart Readymix (#KK5004, Sopachem, Ochten, the Netherlands) for which conditions were optimized per primer pair and are listed, together with the primer sequences, in Appendix A. In total, 21 DNA samples were of sufficient quality to be screened for mutations. Due to technical limitations, we were unable to investigate exon 1 and 8 of *SDHB*, and exon 1 and 2 of *SDHD*. Sanger sequencing was performed as previously described [17]. 

### 4.4. Loss of Heterozygosity

To confirm the loss of the wild type allele in *SDHD*-mutated samples, LOH analysis was performed using two microsatellite markers located upstream and downstream from the dog *SDHD* gene. Primer sequences and PCR conditions are indicated in Appendix A. Furthermore, PCR was performed as previously described [17].

### 4.5. SNP Arrays:

To determine the chromosomal alterations present in dog PCC, we performed SNP arrays on 11 canine PPGLs (Table 2). Canine HD Beadchip SNP arrays (#WG-440-1001, Illumina, San Diego, CA, USA) with a high resolution genome-wide coverage (170,000 SNPs; resolution of approximately 15 SNPs per Mb) were performed and analyzed according to standard procedures at the Human Genomics Facility (HuGeF), Erasmus MC, University Medical Center Rotterdam (www.glimdna.org). 

Final report output files, containing B-allele frequencies and logR ratios, were generated using Illumina BeadStudio Software. The processed files were accordingly visualized using Nexus Copy Number software package (V7.5; Biodiscovery, El Segundo, CA, USA). The SNP array results were submitted to the Gene Expression Omnibus Database. Noisy samples that clearly showed a drop or gain in B-allele frequencies and logR of entire chromosomes were still included in the analysis, taking into account the risk of missing subtle chromosomal changes.

## 5. Conclusions

In conclusion, the results of the current study indicate that similar genomic alterations occur in canine and human PPGLs. Although more functional proof is required to classify the pathogenicity for the SDHD p.Lys122Arg variant, our data suggest that this variant could potentially be pathogenic. Since chromosomal alterations occurred in the dog PPGL at high frequency, affecting chromosomes that are homologous to regions that are also repeatedly lost in human PPGLs, we propose that canine PPGLs are an interesting model for the study of the pathogenesis of human PPGL [22]. More studies are required to identify which common pathways are involved in the pathogenesis of PPGL in both humans and dogs. 

## Figures and Tables

**Figure 1 cancers-11-00607-f001:**
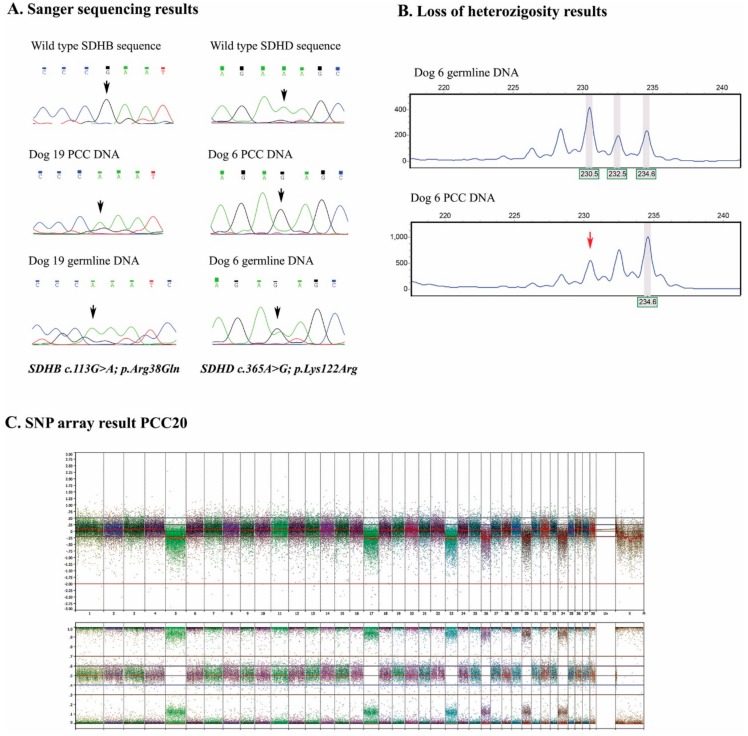
(**A**) In the left panel, succinate dehydrogenase subunit B (*SDHB*) sequences are displayed from healthy reference (upper), PCC19, and corresponding germline DNA. In the right panel, the succinate dehydrogenase subunit D (*SDHD*) sequence is shown from healthy reference, PCC6, and corresponding germline DNA. Note that PCC19 tumor and germline DNA both show the SDHB c.113G>A; p.Arg38Gln variant. PCC6 shows SDHD c.365A>G; p.Lys38Arg in a homozygous fashion in the tumor DNA and heterozygous in the germline DNA, indicating loss of the wild type allele in the tumor. (**B**) Shows loss of heterozygosity of the larger allele in PCC6, confirming the Sanger sequencing results. (**C**) The SNP array result of PCC 20 displays in the upper panel logR ratios, indicating loss of chromosomes 5, 17, 23, 26, 30, and 34. This is also seen in the lower panel by the B-allele frequencies.

**Figure 2 cancers-11-00607-f002:**
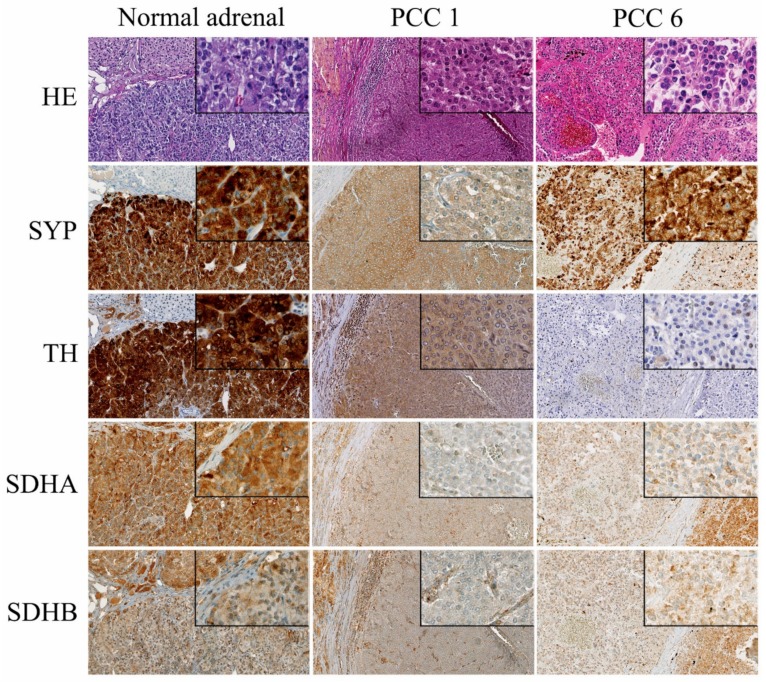
Hematoxylin Eosin staining and immunohistochemistry for synaptophysin (SYP), tyrosine hydroxylase (TH), succinate dehydrogenase subunit A (SDHA) and subunit B (SDHB) of normal dog adrenal, PCC1, and PCC6. Normal adrenal glands were used as positive controls and show strong expression of SYP and tyrosine hydroxylase (TH). PCC1 and 6 label weakly positive for SYP, but for TH only PCC1 shows positive labeling. The normal canine adrenal gland labels positive for SDHA and SDHB, although there is lack of granular labeling, which is characteristic for SDHA and SDHB in human tissues. PCC1 shows labeling of the stromal cells for SDHA and SDHB, which serve as positive control cells, while the PCC cells appear to be heterogeneous weak/negative for SDHB and do not label for SDHA. PCC 6 showed heterogeneous weak expression of SDHB, but there was no difference in labeling intensity between the tumor cells and the normal stromal cells for SDHA. Pictures are at 20× magnification, internal boxes at 40× magnification.

**Figure 3 cancers-11-00607-f003:**
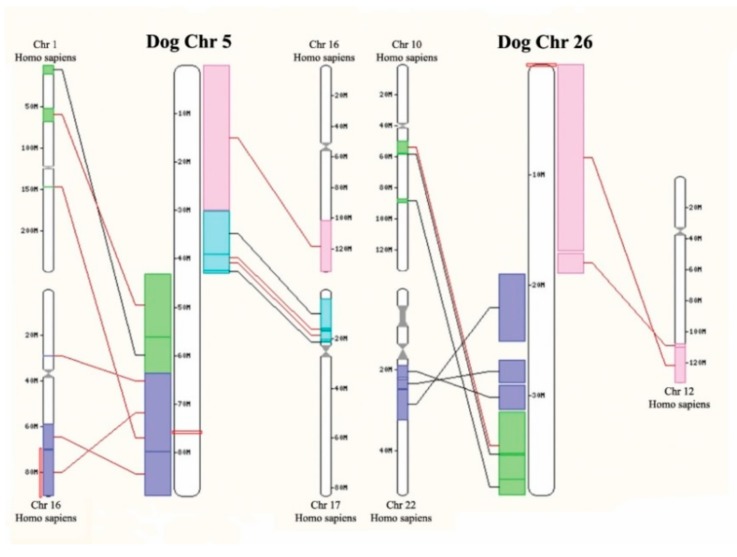
Homology between dog chromosome 5 and 26, and the human genome. It shows the homology between dog chromosome 5 and 26 and several human chromosomes. Of note is that the *SDHD* gene is located on dog chromosome 5 (see Appendix A).

**Table 1 cancers-11-00607-t001:** Variants identified by Sanger Sequencing. SDH-variants that were previously described by Holt et al. [10] as mutations are depicted in bold and italic. If a non-synonymous variation was detected in the tumor, the corresponding germline was investigated for the presence of the variant.

PCC Number	*SDHB* Synonymous	*SDHB* Non-Syn	SDHD Syn	SDHD Non-Syn	Comment
PCC1	p.Y50Y, p.Q164Q, p.A210A		WT		
PCC2	p.Y150Y, p.Q164Q, p.A210A		WT		
PCC3	WT		WT		
PCC4	p.A210A		WT		
PCC5	p.Y150Y, p.Q164Q, p.L188L, p.A210A		WT		
PCC6	p.Y150Y, p.Q164Q, p.A210A		p.A97A	*p.K122R^HO/HE^*	*LOH confirmed*
PCC9	p.Y150Y, p.Q164Q, p.A210A		WT		
PCC10	p.Y150Y, p.Q164Q, p.A210A		WT		
PCC11	p.L188L		WT		
PCC13	p.L188L		WT		
PCC14	p.L188L		WT		
PCC15	p.Y150Y, p.Q164Q, p.L188L, p.A210A		WT		
PCC16	p.L188L		WT		
PCC17	WT		WT		
PCC18	p.L188L		WT		
PCC19	p.Y150Y, p.Q164Q, p.A210A	*p.R38Q^HO/HO^*	WT		*loss Chr. 5*
PCC20	p.L188L		WT		*loss Chr. 5*
PCC21	p.Y150Y, p.Q164Q, p.L188L, p.A210A		WT		
PCC22	p.Y150Y, p.Q164Q, p.L188L, p.A210A		WT		
PCC23	p.Y150Y, p.Q164Q, p.L188L, p.A210A		p.A97A	*p.K122R^HE^*	
PCC46	NA		WT	*p.K122R^HO^*	*loss Chr.5*

HO = homozygous in tumor (PCC46); HE = heterozygous in tumor (PCC23); HO/HE = homozygous in variant in tumor, heterogeneous in germline; HO/HO = homozygous variant in both tumor and germline; LOH = loss of heterozygosity in tumor, loss of chromosome 5 detected with the SNP array are indicated in the comments, non-syn = non-synonymous, syn = synonymous.

**Table 2 cancers-11-00607-t002:** Summary of single nucleotide polymorphism (SNP) array results.

Chromosome	PCC19	PCC20	PCC36 *	PCC37 *	PCC43 *	PCC46 *
2					LOSS	
3					LOSS	
5	LOSS	LOSS	LOSS	LOSS	LOSS	LOSS
7	LOSS			LOSS		LOSS
8					LOSS	
9			GAIN			
12					LOSS	
15					LOSS	
16					LOSS	LOSS
17	LOSS	LOSS				
18					LOSS	
20				LOSS	LOSS	
21					LOSS	
22					LOSS	
23		LOSS				
25					LOSS	
26	LOSS	LOSS	LOSS		LOSS	LOSS
27						
28						
29					LOSS	
30	LOSS	LOSS				
31	LOSS				LOSS	
32	LOSS		LOSS		LOSS	
34	LOSS	LOSS				
35	LOSS				LOSS	

Overview of large chromosomal changes in the informative dog PPGL. * Noisy sample.

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
