# Peer review of "Molecular Alterations in Dog Pheochromocytomas and Paragangliomas"

_cancers, 2019, doi:10.3390/cancers11050607_

Round 1
Reviewer 1 Report
This is a potentially interesting paper about SDHx mutations in canine pheochromocytomas and paragangliomas. However, it is not easy to put the findings together into a clear picture.The study is limited by the fact that variants were only identified in four of the 25 tumors sequenced (2SDHB, 2SDHD), and their pathogenicity is uncertain. A large deletion partly analogous to human chromosomes 1p and 11q was also identified in 5 paragangliomas. While comparable changes are frequent in human PCC/PGL, they are not limited to SDH-deficient tumors. The main problem is that immunohistochemical staining for SDHB is not lost in tumors with either SDHB or SDHD mutations.(Heterogenous staining in one tumor might have resulted from technical issues). It would be helpful to test whether the antibody recognizes canine SDHB in western blots. While positive westerns don't necessarily predict positive IHC, or vice versa, detection of the protein by one or the other method might add clarity.
Author Response
Reviewer 1:
Comments and Suggestions for Authors
“This is a potentially interesting paper about SDHx mutations in canine pheochromocytomas and paragangliomas. However, it is not easy to put the findings together into a clear picture.”
Author reply: The authors acknowledge the reviewers comment that the results are not clearly presented. This has also been mentioned by some other reviewers. Therefore, we have changed the text accordingly throughout the entire document. In addition, we have created a picture with a summary of the results. This is added to the supplemental files.
The study is limited by the fact that variants were only identified in four of the 25 tumors sequenced (2SDHB, 2SDHD), and their pathogenicity is uncertain. A large deletion partly analogous to human chromosomes 1p and 11q was also identified in 5 paragangliomas. While comparable changes are frequent in human PCC/PGL, they are not limited to SDH-deficient tumors. The main problem is that immunohistochemical staining for SDHB is not lost in tumors with either SDHB or SDHD mutations.(Heterogenous staining in one tumor might have resulted from technical issues). It would be helpful to test whether the antibody recognizes canine SDHB in western blots. While positive westerns don't necessarily predict positive IHC, or vice versa, detection of the protein by one or the other method might add clarity.
Author reply: The authors would like to thank the reviewer for the comment about the antibody’s specificity and agree with this point. In addition, we would like to state the following:1) In human PPGL, the presence of an SDHA mutation does not always correspond with a negative SDHA immunohistochemistry. So, although it is a useful tool to detect an SDHA mutation in PPGL, it is far from perfect, even in the human setting. 2) The SDHB immunohistochemistry is more predictive for the presence of an SDH mutation in human PPGL, as most SDH (A, B, C, D and AF2) mutated tumors show an altered, mostly negative staining. However (especially seen in SDHD-mutated human PPGL), we and others have experienced a strong cytoplasmic background staining. Also, some PGL from known SDHD mutation carriers even show a heterogeneous positive staining, most likely due to technical issues (fixation artefacts, tissue damage during the surgery in case of tympanic and jugular PGL). On the contrary, some VHL-mutated tumors also have very low expression of SDHB and are sometimes interpreted as negative by the evaluator.
So, in summary, the SDHA and SDHB immunohistochemistry do not seem to be sufficiently specific for the dog tissues, since the normal dog adrenals did not show the same granular staining as seen in human tissues. In addition, although SDHA and SDHB immunohistochemically negative staining patterns are strong indicators for the presence of an underlying mutation, positive staining cannot exclude the presence of a mutation. Since the non-synonymous SDH variants detected in this study were never detected in human PCC, we cannot draw conclusions about whether the SDH-immunohistochemistry is expected to be negative in the dog PPGL. Therefore, we have diminished/removed parts of text about the SDHA/B immunohistochemistry throughout the entire main document.
Reviewer 2 Report
The authors examine a series of 50 canine paragangliomas and describe germline and tumour events and propose that the dog could have parallels to human forms of inherited paraganglioma.
They first conduct immunohistochemistry for SDHB on the tumours. They then analyze SDHA, B and D using primers.
This study is interesting as it provides a new model of paraganglioma. However, the main impetus for this study is to support that a canine model parallels human germline paragangliomas.
These parallel aspects should be highlighted in the manuscript further. Discussion and diagram of the similar pathogenesis of paragangliomas in dogs vs humans is needed. Homology maps of dog vs human SDHx genes should be also in the main figures, and a map of the variants found in the analysis and how potentially orthologous variants in human SDHx are considered pathogenic or benign. The authors only describe the regions in the supplement.
It is also not clearly delineated what variants are germline and which are tumour based. I would recommend more clarity on this end. Table 1 should clearly state which variants are germline and somatic.
The variants in dog should be annotated as much as possible. Or even human analagous variants should be attempted to be annotated with respect to pathogenicity as per the ACMG guidelines. This should be listed in table 1.
The SNP array analysis should be more integrated into the results of Table 1 and an further detailed explanation on why they are significant and point to LOH in a given SDHx variant. Do these patterns confirm LOH?
Author Response
Reviewer 2
The authors examine a series of 50 canine paragangliomas and describe germline and tumour events and propose that the dog could have parallels to human forms of inherited paraganglioma.
They first conduct immunohistochemistry for SDHB on the tumours. They then analyze SDHA, B and D using primers.
This study is interesting as it provides a new model of paraganglioma. However, the main impetus for this study is to support that a canine model parallels human germline paragangliomas.
These parallel aspects should be highlighted in the manuscript further. Discussion and diagram of the similar pathogenesis of paragangliomas in dogs vs humans is needed. Homology maps of dog vs human SDHx genes should be also in the main figures, and a map of the variants found in the analysis and how potentially orthologous variants in human SDHx are considered pathogenic or benign. The authors only describe the regions in the supplement.
Author reply 1.: The authors acknowledge the fact that the results were not presented in clear fashion and that inappropriately strong statements were made. For clarity we have added a flowchart with a summary of the study including some results. The SNP arrays showed many different chromosomal alterations, but the two most commonly altered chromosomes, dog chromosome 5 and 26 have been highlighted. That is why a figure (Figure 3) has been included in the main document, showing homology with the human chromosomes. The focus of the current manuscript is not to investigate all syntenic regions with altered chromosomes and find it not entirely appropriate to include a whole homology map of dog vs human.
It is also not clearly delineated what variants are germline and which are tumour based. I would recommend more clarity on this end. Table 1 should clearly state which variants are germline and somatic.
Author reply 2: We thank the reviewer for this comment. As mentioned before in author reply 1, is that we agree the study design and results might have not been sufficiently clear, but we have revised the entire manuscript in order for better readability, and have added the flowchart as supplemental Figure 1. In addition, we added a column in table 1 with comments and added an indication of either a “HO/HE” or a “HO/HO” to the non-synonymous variants to indicate if a variant homozygous in the tumor and heterozygous in the germline. We only had germline DNA from 2 dogs (PCC6 and PCC19) that showed the non-synonymous SDH variants. Now mentioned on page 9, line212/213.
The variants in dog should be annotated as much as possible. Or even human analagous variants should be attempted to be annotated with respect to pathogenicity as per the ACMG guidelines. This should be listed in table 1.
Author reply 3: We thank the reviewer for this comment. We have included the right nomenclature and annotation now in the first description of the variants (page 2, last paragraph)
The SNP array analysis should be more integrated into the results of Table 1 and an further detailed explanation on why they are significant and point to LOH in a given SDHx variant. Do these patterns confirm LOH?
Author reply 4: We thank the reviewer for this comment. We have added a column to table 1 and included the chromosomal copy number information in the table. However, not all tumors that were analyzed by SNP arrays were sequenced due to practical reasons. We have now included the information about loss of chromosome 5 in the table, since this might correspond with the loss of the SDHD wild type allele.
Reviewer 3 Report
It is confusing which variants are in the germline and which are somatic. My main questions regard the homozygosity of some of the variants. For example, in PCC6 there is a homozygous variant but also LOH? Is this because was a germline heterozygous mutation, then LOH of the WT allele, and then duplication of the aberrant allele with the mutation in the tumor (i.e. CN-LOH? Also, if there is homozygosity of any of the variants, it would point more likely that these variants are not pathogenic, as usually in PCC/PGL there is a germline heterozygous variant and then a “second hit” in the tumor. Two variants would be a different phenotype, which cause a rare, severe leukodystrophy in humans.Author Response
Comments and Suggestions for Authors
It is confusing which variants are in the germline and which are somatic. My main questions regard the homozygosity of some of the variants. For example, in PCC6 there is a homozygous variant but also LOH? Is this because was a germline heterozygous mutation, then LOH of the WT allele, and then duplication of the aberrant allele with the mutation in the tumor (i.e. CN-LOH?
Author reply: We thank the reviewer for this comment. We have simplified table 1 and added a column with comments, in which we have included the loss of chromosome 5, which was detected by SNP array. Also, we made a notification in the table indicating if the variant was heterozygous or homozygous present in the tumor and germline DNA of the dog. We also indicate now in the materials and methods that germline DNA was only available for PCC6 and PCC19 (page 8, lines 212/213)
Also, if there is homozygosity of any of the variants, it would point more likely that these variants are not pathogenic, as usually in PCC/PGL there is a germline heterozygous variant and then a “second hit” in the tumor. Two variants would be a different phenotype, which cause a rare, severe leukodystrophy in humans.
Author reply: We agree with the reviewer and have changed the statements about the SDHB variant and have weakened those about the SDHD (throughout the entire text). In addition, we have revised the entire document, to improve the readability.
Reviewer 4 Report
Molecular alterations in dog pheochromocytomas and paragangliomas
In their manuscript entitled "Molecular alterations in dog pheochromocytomas and paragangliomas" by Esther Korpershoek and colleagues, the authors investigate a series of 50 canine PPGLs and conclude that canine and human PPGLs show similar genomic alterations, which is suggestive of common interspecies PPGL-related pathways.
While the authors provided an interesting and informative series of canine PPGLs to address the need for an appropriate animal model to study SDH-related PPGL, a few topics might need some further investigation.
1) In total 50 PPGL are included in the study, however from some lesions the diagnosis seems uncertain based on the presented data, this should be clarified:
a. In a few cases no additional immunohistochemical staining or DNA testing is performed (ID24, 34, 45, 48), these might need to be removed from the series as these are not informative for the research question.
b. Few cases were negative for SYP and TH (ID7, 13, 32, 43, 53). Did the authors performed any other staining’s (e.g. chromogranin, S-100) to confirm the diagnosis?
c. Are there any biochemical or radiological data available to confirm the diagnosis?
d. In contract to the result section line 97-98, in one of the cases with negative SYP and TH staining (ID13) gene testing has been performed.
2) Gene analysis showed SDHB and SDHD variants of uncertain significant.
a. The pathogenicity of the identified variants has been discussed in the discussion section but it would be more clear to state the uncertain pathogenicity also in de result section.
3) SNP array analysis was performed in 11 cases of which 9 were analyzed.
a. Abstract line 41-42 in confusing; “Furthermore, the SNP arrays revealed large chromosomal alterations of which chromosome 5, partly homologous to human chromosome 1p, was most frequently lost (100%).” With stated in line 36 that SNP arrays were performed in 11 cases. Loss was present in 6 out of 9 analyzed samples.
b. From the six analyzed samples four showed a noisy result (table 2), reliability should be discussed.
While there is an need for an appropriate animal model to study SDH-related PPGL and the authors preformed a lot of work in these canine PPGLs, the conclusions at this point might be a bit too optimistic. Only few canine PPGLs showed loss of SDH immunohistochemical staining compared to human. Furthermore all identified SDHx variants were of uncertain significant and need further investigation. Therefore the conclusion should be adjusted in a bit more moderate fashion.
Author Response
In their manuscript entitled "Molecular alterations in dog pheochromocytomas and paragangliomas" by Esther Korpershoek and colleagues, the authors investigate a series of 50 canine PPGLs and conclude that canine and human PPGLs show similar genomic alterations, which is suggestive of common interspecies PPGL-related pathways.
While the authors provided an interesting and informative series of canine PPGLs to address the need for an appropriate animal model to study SDH-related PPGL, a few topics might need some further investigation.
1) In total 50 PPGL are included in the study, however from some lesions the diagnosis seems uncertain based on the presented data, this should be clarified: a. In a few cases no additional immunohistochemical staining or DNA testing is performed (ID24, 34, 45, 48), these might need to be removed from the series as these are not informative for the research question.
1. Author reply: We would like to thank the reviewer for this comment. However, since this is a quite unbiased series, the clinical findings (such as frequencies of tumor type per gender etc) might be of interest and would change if we would remove them from the case series.
b. Few cases were negative for SYP and TH (ID7, 13, 32, 43, 53). Did the authors performed any other staining’s (e.g. chromogranin, S-100) to confirm the diagnosis?
b. Author reply: We agree that not all the samples that were sequenced were SYP or TH positive. We have performed these immunohistochemical stainings with human antibodies, but the antibodies appeared to be human specific and did not result in a positive staining in the dog tissues.
c. Are there any biochemical or radiological data available to confirm the diagnosis?
c. Author reply: The diagnosis PCC was mostly done based on HE stainings after autopsy. I do not have this data and since my collaborator from Bristol is now working in Australia, it might be difficult to collect.
d. In contract to the result section line 97-98, in one of the cases with negative SYP and TH staining (ID13) gene testing has been performed.
d. Author reply: The author acknowledges this fact. This case was in the first screening series and did not stain positive for TH or SYP. Since we already started with the Sanger sequencing, we added it to the series anyway. The diagnosis was also performed by a pathologist based on the HE staining. We have accordingly added that this case was included in the sanger series of 25 PPGL on page 9, line 238
2) Gene analysis showed SDHB and SDHD variants of uncertain significant. a. The pathogenicity of the identified variants has been discussed in the discussion section but it would be more clear to state the uncertain pathogenicity also in de result section.
Author reply 2: We agree with the reviewer and have changed the statements about the SDHB variant and have weakened those about the SDHD (throughout the entire text). In addition, we have revised the entire document, to improve the readability.
3) SNP array analysis was performed in 11 cases of which 9 were analyzed. a. Abstract line 41-42 in confusing; “Furthermore, the SNP arrays revealed large chromosomal alterations of which chromosome 5, partly homologous to human chromosome 1p, was most frequently lost (100%).” With stated in line 36 that SNP arrays were performed in 11 cases. Loss was present in 6 out of 9 analyzed samples.
Author reply 3: We thank the reviewer for this comment, but want to mention that the 100% refers to the presence of loss of chromosome 5 in samples showing chromosomal alterations. We have now changed this in the abstract (page 1, lines 41-42). In the results section, this was already mentioned “100% of 6 dog PPGLs”, after first mentioning why the additional 3 were left out of this figure.
b. From the six analyzed samples four showed a noisy result (table 2), reliability should be discussed.
b. Author reply: We agree that the reliability of the noisy samples should be mentioned. We have now accordingly mentioned this in the materials and methods section. Page 9, line 267-269) “Noisy samples that clearly showed a drop or gain in B-allele frequencies and logR of entire chromosomes were still included in the analysis, taking into account the risk of missing subtle chromosomal changes.”
While there is an need for an appropriate animal model to study SDH-related PPGL and the authors preformed a lot of work in these canine PPGLs, the conclusions at this point might be a bit too optimistic. Only few canine PPGLs showed loss of SDH immunohistochemical staining compared to human. Furthermore all identified SDHx variants were of uncertain significant and need further investigation. Therefore the conclusion should be adjusted in a bit more moderate fashion.
Author reply: We acknowledge the fact that our statements were too strong, and that the immunohistochemistry results were not appropriate. Therefore, we have revised the entire document in order to tune down the statements, but also to improve the readability.
Round 2
Reviewer 1 Report
My previous comments have been adequately addressed
Author Response
My previous comments have been adequately addressed
Reply: We thank the reviewer for this statement.
Reviewer 2 Report
The authors have addressed some of my suggestions. It is still not clear tumour vs germline analysis and it should be further explained in the new flowchart. It seems most of the genetic analysis was done in the tumour, and likely allele fraction could not be used as an indicator of a germline event.
The relevant genes on the human and dog chromosomes should be indicated in figure 3 (e.g. SDH genes). There are some grammatical errors and should be copy written (e.g. heterozygous and not heterogeneous).
Author Response
The authors have addressed some of my suggestions. It is still not clear tumour vs germline analysis and it should be further explained in the new flowchart. It seems most of the genetic analysis was done in the tumour, and likely allele fraction could not be used as an indicator of a germline event.
Author reply: We thank the reviewer for this comment. Since it appears that it was not clearly stated that we only investigated tumors and only if a non-synonymous variant was identified in the tumor, we accordingly investigated the germline DNA (if available), we have added/removed the following sentences to the text:
Page 1; line 37: “all normal dog adrenals and” was removed from the abstract, since these normal adrenals were only used as positive controls for the immunohistochemistry staining. This probably is why it was confusing to understand if tumor or germline was used in this study. Hopefully it is now clear we only looked at PPGLs and in a few cases also in the germline.
Page 1; line 38: “… of the canine tumors…” was added
Page 2, line 88-89: “If a tumor presented a non-synonymous variant, the presence of this variant was also investigated in corresponding germline DNA.” was added.
Page 2, line 92: “…was only available for PCC6 ...” was added.
Page 6, Table 1 (legends), line 148-150: “If a non-synonymous variation was detected in the tumor, corresponding germline was investigated for the presence of the variant.” was added.
Supplemental figure 1: The results of the sanger sequencing and LOH/SNP array have been added to the flowchart, as well as the number of PPGL in the top box.
The relevant genes on the human and dog chromosomes should be indicated in figure 3 (e.g. SDH genes). There are some grammatical errors and should be copy written (e.g. heterozygous and not heterogeneous).
Author reply: We thank the reviewer for this comment. We have added a supplemental figure 2 indicating the genomic locations of SDHA, SDHB, SDHC and SDHD in the dog genome. We have also mentioned the SDHD location on dog chromosome 5 in de legends of Figure 3, referring to the supplemental figure.
In addition, one of the coauthors, professor Day (a native speaker), has edited the text. He has made changed throughout the entire text, which is indicated with ‘track changes’ Hopefully it now meets the standards for publication.
Reviewer 3 Report
Appreciate clarifications- would further clarify in the abstract and results that most of the analysis are in the tumor without germline available, since SDH germline mutations are of such interest in human paragangliomas
Author Response
Appreciate clarifications- would further clarify in the abstract and results that most of the analysis are in the tumor without germline available, since SDH germline mutations are of such interest in human paragangliomas
Reply: We thank the reviewer for this comment, and have accordingly removed the sentence below from the abstract, since this might have caused the impression that normal adrenals were also investigated in this study. We only used normal adrenals as positive controls for the immunohistochemistry. In addition, we have attempted to clarify the use of tumors and in some cases corresponding germline DNA by adding additional sentences throughout the text.
Page 1; line 37: “all normal dog adrenals and” was removed.
Page 1; line 38: “… of the canine tumors…” was added
Page 2, line 88-89: “If a tumor presented a non-synonymous variant, the presence of this variant was also investigated in corresponding germline DNA.” was added.
Page 2, line 92: “…was only available for PCC6 ...” was added.
Page 6, Table 1 (legends), line 148-150: “If a non-synonymous variation was detected in the tumor, corresponding germline was investigated for the presence of the variant.” was added.
In addition, we would like to state that only PPGL (and not normal adrenals) are now mentioned in the abstract, making it more clear that tumors were the main focus of the study. We agree with the reviewer that germline mutations are of vast interest, but conceptually somatic mutations are also of interest, since this would also provide some “indications” that the SDHD p.K122R variant could be pathogenic. Although we have not identified somatic variants as yet, the previous study on dog PCC did find the SDHD variant in a somatic fashion. And we did find this variant in 3 tumors (one germline).
Reviewer 4 Report
Many thanks for the revised version of the manuscript in order to tune down the statements. However, there might still be some room to improve the readability.
1) in particular to discriminate heterozygous and homozygous in relation to germline and tumor testing in Table 1. The legend is confusing according to samples in which no germline testing has been performed (PCC23/46) and ‘in variant’ should be removed after HO/HE.
2) the terminology could be more consistent throughout the manuscript (for example DNA variants vs alteration vs mutation) and the used variant classification should be mentioned in the method. The term potential pathogenic is unclear.
3) supplemental figure 1 in this form might implied parallel analysis of 50, 11 and 25 PPGL, while all used (molecular) methods are applied to different subgroups of one cohort, this could be more clear in a revised flowchart.
4) Minor detail; some gene names are not in italic (for example line 166).
Author Response
Many thanks for the revised version of the manuscript in order to tune down the statements. However, there might still be some room to improve the readability.
Reply: We thank the reviewer for these comments and hope to have changed the text and figures according the reviewers suggestions.
1) in particular to discriminate heterozygous and homozygous in relation to germline and tumor testing in Table 1. The legend is confusing according to samples in which no germline testing has been performed (PCC23/46) and ‘in variant’ should be removed after HO/HE.
Reply 1: We have removed ‘in variant’ from the legends. (Page 6, line 161) In addition, “If a non-synonymous variation was detected in the tumor, corresponding germline was investigated for the presence of the variant.” was added (page 6, line 158-160).
“homozygous in tumor (PCC46); HE = heterozygous in tumor (PCC23);”(page 6, line 161-162) was also added to the legends.
2) the terminology could be more consistent throughout the manuscript (for example DNA variants vs alteration vs mutation) and the used variant classification should be mentioned in the method. The term potential pathogenic is unclear.
Reply 2: We have now changed “variant-alteration-mutation” according the right terminology.
(page 1, line 29) “Mutations” was replaced with “genetic alterations”
(Page 2, line 69) potential pathogenic variants” was replaced with “genetic variants that might be potentially pathogenic”
(Page 2, line 70) “mutation” was removed, and the second “mutation” was changed into “alterations”.
(Page 7, line 176-177) “mutation” was replaced with “genetic alterations”
(Page 7, line 179) “variants” was replaced with “genetic alterations”
(page 10, line 296) “Mutations” was replaced with “alterations”
3) supplemental figure 1 in this form might implied parallel analysis of 50, 11 and 25 PPGL, while all used (molecular) methods are applied to different subgroups of one cohort, this could be more clear in a revised flowchart.
Reply 3: This has been changed in supplemental figure 1.
4) Minor detail; some gene names are not in italic (for example line 166).
Reply 4: We have changed gene names in italic throughout the text (if referring to DNA, not an amino-acid).